# Tmem263 deletion disrupts the GH/IGF-1 axis and causes dwarfism and impairs skeletal acquisition

Dylan C Sarver[1], Jean Garcia-Diaz[2,3,4], Muzna Saqib[1], Ryan C Riddle[2,3,5], G William Wong[1]*

[1]Department of Physiology, Johns Hopkins University School of Medicine, Baltimore, United States; [2]Department of Orthopaedic Surgery, Johns Hopkins University School of Medicine, Baltimore, United States; [3]Department of Orthopaedics, University of Maryland School of Medicine, Baltimore, United States; [4]Cell and Molecular Medicine graduate program, Johns Hopkins University School of Medicine, Baltimore, United States; [5]Research and Development Service, Baltimore Veterans Administration Medical Center, Baltimore, United States

*For correspondence:
gwwong@jhmi.edu

**Abstract** Genome-wide association studies (GWAS) have identified a large number of candidate genes believed to affect longitudinal bone growth and bone mass. One of these candidate genes, *TMEM263*, encodes a poorly characterized plasma membrane protein. Single nucleotide polymorphisms in TMEM263 are associated with bone mineral density in humans and mutations are associated with dwarfism in chicken and severe skeletal dysplasia in at least one human fetus. Whether this genotype-phenotype relationship is causal, however, remains unclear. Here, we determine whether and how TMEM263 is required for postnatal growth. Deletion of the *Tmem263* gene in mice causes severe postnatal growth failure, proportional dwarfism, and impaired skeletal acquisition. Mice lacking Tmem263 show no differences in body weight within the first 2 weeks of postnatal life. However, by P21 there is a dramatic growth deficit due to a disrupted growth hormone (GH)/insulin-like growth factor 1 (IGF-1) axis, which is critical for longitudinal bone growth. *Tmem263*-null mice have low circulating IGF-1 levels and pronounced reductions in bone mass and growth plate length. The low serum IGF-1 in *Tmem263*-null mice is associated with reduced hepatic GH receptor (GHR) expression and GH-induced JAK2/STAT5 signaling. A deficit in GH signaling dramatically alters GH-regulated genes and feminizes the liver transcriptome of Tmem263-null male mice, with their expression profile resembling wild-type female, hypophysectomized male, and Stat5b-null male mice. Collectively, our data validates the causal role for Tmem263 in regulating postnatal growth and raises the possibility that rare mutations or variants of *TMEM263* may potentially cause GH insensitivity and impair linear growth.

## eLife assessment

This study discloses **important** physiological function for TMEM63 in regulating postnatal growth in mice. The data supporting the impaired body growth and skeletal phenotype as well as disrupted growth hormone/insulin-like growth factor-I (GH/IGF-I) signaling in TMEM63 knockout mice are **compelling**. However, to establish that alteration of hepatic GH/IGF-I signaling is the cause for observed growth and skeletal phenotype in TMEM63 knockout mice would need additional work.

## Introduction

Skeletal development, growth, and the maintenance of tissue structure and mass across the lifespan are regulated by complex interactions between genetics and environment. A better understanding of these interactions and how they influence the function of cells responsible for skeletal homeostasis will provide actionable targets to mitigate skeletal disease. In exploring the heritability of bone growth and mass, genome-wide association studies (GWAS) have identified a large number of potential genes associated with bone mineral density (BMD) (*Estrada et al., 2012*; *Kemp et al., 2017*), bone structure (*Zhao et al., 2010*; *Styrkarsdottir et al., 2019*), and height (*Chan et al., 2015*). However, it remains a daunting challenge to prioritize and establish the causal role for these candidate genes. Consequently, only a limited number of the GWAS candidate genes have been functionally validated to play a causal role.

TMEM263, also referred to as C12orf23, was identified through associations between femoral neck BMD and heel bone BMD (*Estrada et al., 2012*; *Kemp et al., 2017*). Its function is unknown, but subsequent gene network analysis placed TMEM263 in a module related to osteoblast function (*Calabrese et al., 2017*). Intriguingly, a nonsense mutation (Trp59*) that truncates the TMEM263 protein is linked to dwarfism in chicken (*Wu et al., 2018*), although it is unclear if this mutation directly causes dwarfism. More recently, a two nucleotide deletion that causes a frameshift and premature termination in *TMEM263* was implicated as a candidate gene for a severe case of autosomal-recessive skeletal dysplasia in a fetus (*Mohajeri et al., 2021*). Apart from this limited information, little is known about TMEM263 and its function.

It is known that the growth hormone (GH) and insulin-like growth factor 1 (IGF-1) axis plays a critical and essential role in postnatal growth (*David et al., 2011*; *Efstratiadis, 1998*; *Savage et al., 2011*; *Rosenfeld et al., 2007*; *Rosenfeld et al., 1994*; *Qian et al., 2022*). Defect in any component of this axis causes a variable spectrum (modest to severe) of growth retardation and skeletal dysplasia in humans and in animal models (*Lupu et al., 2001*; *Zhou et al., 1997*; *Baker et al., 1993*; *Liu et al., 1993*; *Davey et al., 1999b*; *Kofoed et al., 2003*; *Udy et al., 1997*; *Hwa et al., 2007*; *Klammt et al., 2018*; *Godowski et al., 1989*; *Woods et al., 1996*; *Abuzzahab et al., 2003*; *Lanning and Carter-Su, 2006*). Since mutation in *TMEM263* is linked to human skeletal dysplasia and dwarfism in chicken, and *TMEM263* is also a GWAS candidate gene for BMD, we speculated that TMEM263 may be a novel regulator of the GH/IGF-1 axis. We therefore set out to test this hypothesis using a genetic approach.

We showed here that Tmem263 is not required for embryonic development, prenatal growth, and fertility. However, mice lacking Tmem263 exhibit growth arrest and have severe reduction in bone mass and growth plate length. We further showed that these defects are due to a marked reduction in GH receptor (Ghr) mRNA and protein expression in the liver. This resulted in greatly diminished GH-induced JAK2/STAT5 signaling critical for IGF-1 synthesis and secretion by the liver. Consequently, *Tmem263* knockout (KO) mice had low circulating IGF-1 and IGF binding protein 3 (IGFBP3), leading to dramatic postnatal growth failure and skeletal dysplasia. Our loss-of-function studies have provided direct evidence that Tmem263 is a causal gene affecting BMD and postnatal growth, and it does so by regulating the GH/IGF-1 axis, specifically hepatic GHR protein levels. The GH insensitivity (GHI) phenotype seen in *Tmem263*-KO mice raises the intriguing possibility that rare *TMEM263* mutations or variants may potentially disrupt postnatal linear growth due to deficiency in GHR signaling.

## Results

### TMEM263 is a conserved and widely expressed plasma membrane protein

The human *TMEM263* gene encodes a small protein of 116 amino acids with two predicted transmembrane helices (*Figure 1A*). It is highly conserved across vertebrate species, with the fish (*Danio rerio*), frog (*Xenopus tropicalis*), chicken (*Gallus gallus*), and mouse (*Mus musculus*) orthologs sharing 73, 77, 90, and 97% amino acid identity with the full-length human TMEM263. In humans, the *TMEM263* transcript is widely and variably expressed across tissues (*Figure 1B*), with liver having the highest expression. The expression data is based on the consensus dataset of normalized expression (nTPM) levels across tissues by combining the Human Protein Atlas (HPA) (*Uhlén et al., 2015*) and Gene-Tissue Expression (GTEx) transcriptomics datasets (*The Gtex Consortium, 2015*). Reflecting the transcript expression profile, human TMEM263 protein is also widely expressed (https://www.proteinatlas.

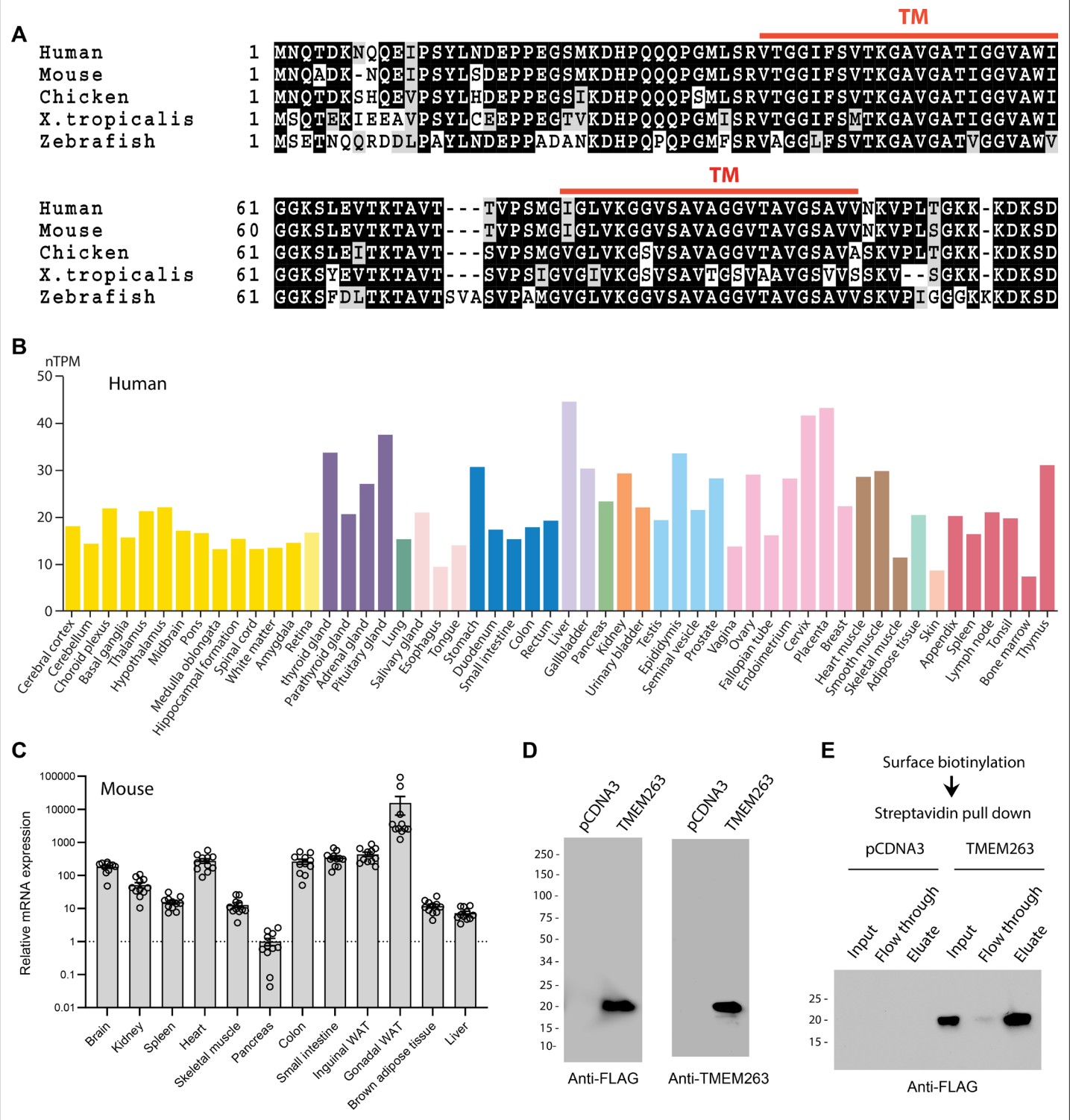

**Figure 1.** TMEM263 is a novel and highly conserved plasma membrane protein. (**A**) Sequence alignment of full-length human (NP_689474), mouse (NP_001013046), chicken (NP_001006244), *Xenopus* frog (NP_989399), and zebrafish (NP_998306) transmembrane protein 263 (TMEM263) using Clustal-omega (**Sievers and Higgins, 2018**). Identical amino acids are shaded black and similar amino acids are shaded gray. Gaps are indicated by dash lines. The two predicted transmembrane domains are indicated in red. (**B**) *TMEM263* expression across normal human tissues based on the consensus Human Protein Atlas (HPA) and Gene-Tissue Expression (GTEx) datasets. The data can be accessed via the HPA database (https://www.proteinatlas.org). nTPM denotes normalized protein-coding transcripts per million and it corresponds to the mean values of the different individual samples from each tissue. Bars are color-coded based on tissue groups with functional features in common. (**C**) Tmem263 expression across mouse tissues (n=11). Relative

*Figure 1 continued on next page*

*Figure 1 continued*

expression across tissues were first normalized to *β-actin*, then normalize to the tissue (pancreas) with the lowest expression. (**D**) Immunoblot analysis of cell lysate from HEK293 cells transfected with a control pCDNA3 empty plasmid or plasmid encoding human TMEM263 tagged with a C-terminal Myc-DDK epitope. Immunoblots were probed with an anti-FLAG (DDK) antibody (left panel) or an anti-TMEM263 antibody (right panel). (**E**) TMEM263 is localized to the plasma membrane. Surface biotinylation was carried out on transfected HEK293 cells. Biotinylated plasma membrane proteins were captured with Avidin-agarose beads, eluted, and immunoblotted for TMEM263 with an anti-FLAG antibody.

The online version of this article includes the following source data for figure 1:

**Source data 1.** Top left – Original uncropped membrane from imager showing blue channel only as a black and white image.

**Source data 2.** Left – Original uncropped membrane from imager showing blue channel only as a black and white image.

org) (*Uhlén et al., 2015*). The mouse *Tmem263* is also variably and widely expressed across tissues (*Figure 1C*).

Human TMEM263 protein is ~20 kDa in size when expressed in mammalian HEK293 cells (*Figure 1D*). The presence of two putative hydrophobic segments suggests that TMEM263 may be an integral membrane protein. However, different transmembrane prediction programs have given inconsistent results. To experimentally determine whether human TMEM263 is a membrane protein localized to the cell surface or other intracellular membrane compartments, we performed surface biotinylation followed by streptavidin pull-down and immunoblot analysis. The cell-impermeable biotinylation reagent labeled only proteins on the surface of intact cells. This analysis showed that TMEM263 is localized to the plasma membrane when expressed in HEK293 cells (*Figure 1E*).

## Tmem263 is not required for development and prenatal growth

A genetic approach was used to determine the in vivo function of Tmem263. The mouse *Tmem263* gene (NM_001013028) consists of three exons, and exon 3 encodes ~81% of the full-length protein. To generate a loss-of-function mouse model, we used CRISPR/Cas9 method to delete the entire protein-coding portion of exon 3, thus ensuring a complete null allele (*Figure 2A*). We performed genomic PCR and DNA sequencing to confirm the genotype and deletion of the *Tmem263* gene (*Figure 2A–B*). As expected, deletion of the *Tmem263* gene resulted in complete absence of the transcript in liver and hypothalamus (*Figure 2C*).

The genotype distribution of 82 pups at postnatal day 1 (P1) largely conformed to the expected Mendelian ratio for wild-type (WT) (+/+), heterozygous (+/-), and KO (-/-) mice (*Figure 2D*), indicating that Tmem263 is not required for embryonic development. Loss of Tmem263 did not affect suckling at P1, as the milk spot was clearly visible in *Tmem263*-KO pups (*Figure 2E*). No gross anatomical abnormalities in the axial skeleton, appendages, and cartilage were noted in *Tmem263*-KO pups at P1, indicating that Tmem263-KO mice were born normal (*Figure 2F*). The birth weights of pups at P1 were also not different between genotypes (*Figure 2G*). Together, these data indicate that Tmem263 is not required for embryonic development and prenatal growth.

## Tmem263 deletion causes severe postnatal growth retardation

From P1 to P14, the weights of *Tmem263*-KO pups were not different from WT controls (*Figure 2G*), indicating no deficit in growth trajectory within the first 2 weeks of life. However, before and around the time of weaning at P21 when mice started to experience an accelerated growth spurt, coincident with the initiation of GH action (*Lupu et al., 2001*), clear differences in body weight were noted in *Tmem263*-KO mice relative to WT (+/+) and heterozygous (+/-) controls (*Figure 2G*). By adulthood (8 weeks of age), a severe growth failure phenotype was apparent in the *Tmem263*-KO mice (*Figure 2H*).

Both male and female KO mice exhibited the same striking degree of dwarfism, with body weight and body length dramatically reduced relative to WT controls (*Figure 2I–K*). Due to their small size, the organ weights (liver, pancreas, heart, kidney, spleen, gastrocnemius, and brain) of the KO mice were also significantly lower relative to WT controls (*Figure 2—figure supplement 1*). When normalized to their body weight, *Tmem263*-KO mice also had lower relative weights of liver (male only), spleen (male only), and skeletal muscle (gastrocnemius) and higher kidney and brain weight (*Figure 2—figure supplement 1*). Partial loss of *Tmem263* did not result in haploid insufficiency, as the heterozygous (+/-) mice were indistinguishable from WT littermates across all measured

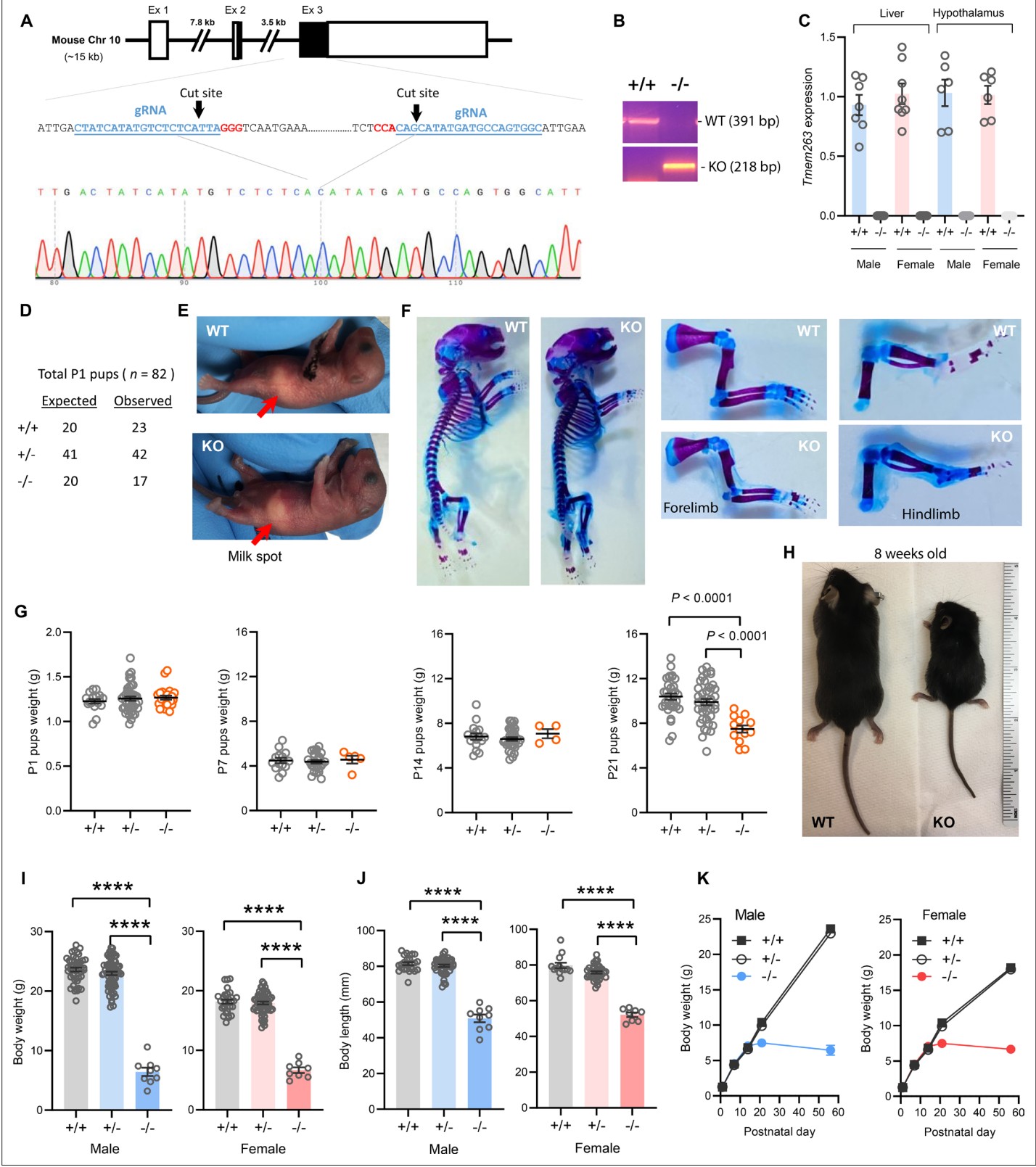

**Figure 2.** Deletion of *Tmem263* gene causes dwarfism. (**A**) Generation of *Tmem263* knockout (KO) mice. The exon 3 that encodes ~81% of the full-length protein was deleted using CRISPR/Cas9 method and confirmed with DNA sequencing. The location and sequence of the two guide RNAs (gRNA) used to generate the deletion were underlined. Filled-in black boxes indicate part of the exon that codes for Tmem263 protein, and white boxes indicate part of the exon that codes for 5' and 3' UTR of the transcript. (**B**) Wild-type (WT) and KO alleles were confirmed by PCR genotyping.

*Figure 2 continued on next page*

*Figure 2 continued*

(**C**) The complete loss of *Tmem263* transcript in KO mice was confirmed by qPCR in male and female mouse liver and hypothalamus (WT, n=6–8; KO, n=6–8). (**D**) The expected Mendelian versus observed genotype distributions in postnatal day 1 (P1) pups (n=82). (**E**) Representative images of WT and *Tmem263*-KO pups at **P1**. Milk spots are indicated by a red arrow. (**F**) Representative Alcian blue and Alizarin red staining of axial skeletal and cartilage in WT and KO P1 pups. (**G**) Body weights of WT and *Tmem263*-KO pups at P1 (WT = 17; Het = 42; KO = 23), P7 (WT = 15; het = 28; KO = 5), P14 (WT = 16; het = 34; KO = 4), and P21 (WT = 33; het = 41; KO = 13). For panel G, we combined the data of male and female pups from P1 to P21. (**H**) Representative images of adult WT and *Tmem263*-KO mice at 9 weeks of age. (**I–J**) Body weights and body length of WT (+/+), heterozygous (+/-), and KO (-/-) male and female mice at 9 weeks of age. Sample size for males (WT = 45; het = 73; KO = 9) and females (WT = 30; het = 59; KO = 8). (**K**) The growth curve trajectory based on the combined data in G and I. All data are presented as mean ± SEM. ****p<0.0001 (one-way ANOVA with Tukey's multiple comparisons test).

The online version of this article includes the following figure supplement(s) for figure 2:

**Figure supplement 1.** Absolute and relative tissue and organ weights of *Tmem263* wild-type (WT) (+/+), heterozygous (+/-), and knockout (KO) (-/-) male and female mice.

parameters. Both male and female *Tmem263*-KO mice were fertile, as indicated by viable mouse litters derived from multiple independent matings between male (-/-) and female (-/-) mice. The litter size in general was ~30–40% smaller, though this observation was based only on a limited number of litters produced. The smaller litter size might reflect potential constrains imposed by the size of the KO dam. Mice born from homozygous KO (-/-) parents appeared largely normal with no gross physical or behavioral abnormalities noted except their pronounced dwarfism. Together, these data indicate that Tmem263 is required for postnatal growth and its absence causes postnatal growth retardation.

## Pronounced skeletal dysplasia in Tmem263-null mice

In accordance with postnatal growth failure, adult *Tmem263*-KO mice had much shorter femoral bones compared to WT and heterozygous controls (*Figure 3A–B*). Quantification of various bone parameters in the distal femur indicated a marked reduction in trabecular bone volume, trabecular number, trabecular bone thickness, cortical tissue area, and cortical thickness in *Tmem263*-KO relative to WT and heterozygous mice (*Figure 3C–H*, regions of interest [ROI] for both trabecular and cortical bone analysis were adjusted in proportion to the length of bone, see Materials and methods). However, normalization of cortical tissue area and cortical thickness to femur length indicated that skeletal geometry was proportional to that in WT littermates (*Figure 3I–J*). Histological analysis revealed a much thinner growth plate in *Tmem263*-KO mice (*Figure 3K*). Examination of growth plate morphology in the proximal tibia revealed a reduction in total growth plate length that was secondary to reductions in proliferative zone length in *Tmem263*-KO mice relative to WT controls (*Figure 3L–M*). The hypertrophic zone length was not different between genotypes (*Figure 3N*). Together, these data indicate that Tmem263 deficiency severely affects postnatal skeletal growth and acquisition.

## Tmem263-null mice have low IGF-1, IGFBP3, and IGFALS levels

The GH/IGF-1 axis plays an essential role in postnatal growth and its deficiency results in growth failure (*David et al., 2011*; *Savage et al., 2011*; *Hwa et al., 2021*). The growth retardation and skeletal dysplasia phenotypes seen in *Tmem263*-KO mice strongly suggest a potential deficit in the GH/IGF-1 axis. *Tmem263*-KO mice had a marginal, though not significant, increase in serum GH compared to WT controls (*Figure 4A*). However, serum IGF-1, IGFBP3, and IGF acid labile subunit (IGFALS) levels were markedly lower in *Tmem263*-KO mice relative to WT controls (*Figure 4B–D*). Serum insulin levels were significantly lower in KO female mice and trended lower in KO male mice relative to WT controls (*Figure 4E*). Random-fed blood glucose levels, however, were significantly lower in *Tmem263*-KO mice (*Figure 4F*). The combination of lower insulin and blood glucose levels suggests enhanced insulin sensitivity. These clinical features are also commonly seen in patients with GHR mutations (*Savage et al., 2011*). Given the bone phenotype seen in the *Tmem263*-KO mice, we also measured serum $Ca^{2+}$ and phosphate levels. While serum $Ca^{2+}$ levels were reduced in Tmem263-KO mice, serum phosphate levels and the $Ca^{2+}$/P ratio were not different between genotypes (*Figure 4G–I*). Together, these data indicate that low IGF-1, IGFBP3, and IGFALS levels likely contribute to growth retardation and skeletal dysplasia in *Tmem263*-KO mice.

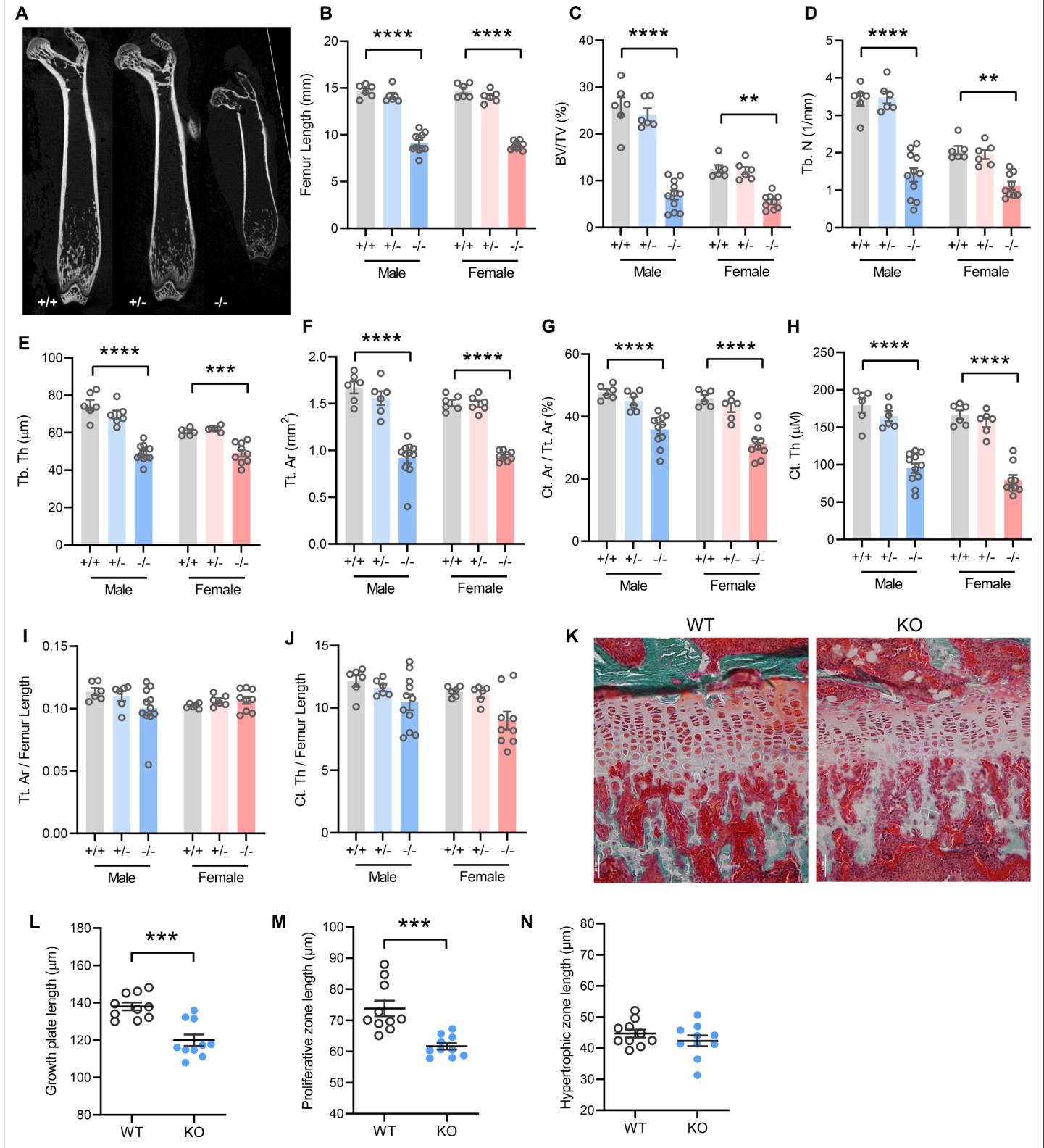

**Figure 3.** Mice lacking TMEM263 exhibit pronounced skeletal dysplasia. (**A**) Representative microCT images of bone (femur) showing a dramatic reduction in size in *Tmem263* knockout (KO) (-/-) mice relative to wild-type (WT) (+/+) and heterozygous (+/-) controls at 8 weeks of age. (**B**) Femur length of WT (+/+), heterozygous (+/-), and KO (-/-) male and female mice at 8 weeks of age. (**C**) Quantification of trabecular bone volume per tissue volume (BV/TV) in the distal femur of WT (+/+), heterozygous (+/-), and KO (-/-) male and female mice. (**D**) Quantification of trabecular number (Tb. N) in

*Figure 3 continued on next page*

Figure 3 continued

the distal femur. (**E**) Trabecular bone thickness (Tb. Th). (**F**) Cortical tissue area (Tt. Ar). (**G**) Cortical area per tissue area. (**H**) Cortical thickness. (**I**) Tissue area per femur length. (**J**) Cortical thickness per femur length in male and female mice. (**K**) Representative images of tibial growth plate histology in WT and *Tmem263*-KO male mice. (**L–N**) Quantification of growth plate length (**L**), proliferative zone length (**M**), and hypertrophic zone length (**N**) in WT (n=10) and KO (n=10) male mice. All data were collected on 8-week-old mice. Sample size for panels B–J: males (WT = 6; het = 6; KO = 11) and females (WT = 6; het = 6; KO = 9). All data are mean ± SE. **p<0.01; ***p<0.001; ****p<0.0001 (one-way ANOVA with Tukey's multiple comparisons test).

## Tmem263-null mice have a marked deficit in hepatic GHR expression and signaling

The GH/IGF-1 axis (*Figure 5A*) required for postnatal growth consists of (i) hypothalamic GH releasing hormone which induces pituitary GH release; (ii) secreted GH then binds to and activates GHR in the liver, bone, and other peripheral tissues to induce the synthesis and secretion of IGF-1; (iii) the direct effects of GH on bone, together with the endocrine and autocrine/paracrine actions of IGF-1, promote longitudinal bone growth (*Lupu et al., 2001*; *Ohlsson et al., 2009*; *Stratikopoulos et al., 2008*; *Wu et al., 2009*). A majority (~75%) of the circulating IGF-1 is secreted by the liver (*Ohlsson et al., 2009*). Because *Tmem263*-KO mice had normal to marginally higher GH but low IGF-1 and IGFBP3 levels, the GH/IGF-1 axis is likely disrupted at the liver. Indeed, the expression of *Ghr*, *Igf1*, and *Igfals* (IGF binding protein acid labile subunit) transcripts was significantly reduced (~4-fold) in the liver of *Tmem263*-KO mice relative to WT controls (*Figure 5B*). The expression of *Igfbp3*, however, was not different between genotypes. Consistent with the mRNA data, the hepatic GHR protein level was similarly and greatly reduced in *Tmem263*-KO mice compared to WT controls (*Figure 5C–D*).

Reduced Ghr transcript and protein expression would diminish the magnitude of GH-induced signaling, resulting in GHI. To test this, mice were injected with recombinant GH and GH-induced signaling in liver was assessed. Janus kinase 2 (JAK2) and signal transducer and activator of transcription 5b (STAT5b) phosphorylation are the key essential signaling events downstream of the GHR (*Lanning and Carter-Su, 2006*; *Rosenfeld and Hwa, 2009*). At baseline when mice were not injected with GH, there were no detectable phospho-Jak2 and phopho-Stat5b signals (*Figure 5E*). In contrast, WT male mice injected with GH showed a robust GH-stimulated phosphorylation of Jak2 and Stat5b (*Figure 5F*). Relative to WT controls, GH injection elicited a greatly diminished response in *Tmem263*-KO mice, as indicated by the much lower magnitude (~60–80% reduction) of Jak2 and Stat5b phosphorylation relative to WT controls (*Figure 5F–G*). Together, these data point to reduced hepatic GHR expression and GHI as potentially contributing to growth failure and skeletal dysplasia seen in the *Tmem263*-KO animals.

## Loss of Tmem263 dramatically affects GH-regulated genes in the liver

Based on our current data and what is known in the literature (*Clodfelter et al., 2006*; *Holloway et al., 2007*), impaired GH signaling results in widespread changes in the GH-regulated liver transcriptome of *Tmem263*-KO male mice. Indeed, 8.6% of the liver transcriptome in *Tmem263*-KO male mice were significantly altered relative to WT controls, with 1547 protein-coding genes up-regulated and 862 protein-coding genes down-regulated (*Figure 6A*). Among the major classes of genes and pathways most affected by the loss of Tmem263 are those that involved in lipid and steroid metabolism, detoxification process, and xenobiotic handling (*Figure 6B*); all of the affected genes and pathways are known to be markedly altered in male mice in which GH signaling is disrupted due to Stat5b deficiency or an altered GH secretory profile (i.e. loss of pulsatile release of GH) (*Udy et al., 1997*; *Clodfelter et al., 2006*; *Holloway et al., 2007*; *Davey et al., 1999a*; *Holloway et al., 2006*; *Lau-Corona et al., 2017*; *Waxman et al., 1991*).

It is well known that GH and its distinct secretory profiles (continuous in females vs. intermittent in males) plays a critical role in establishing and maintaining sexually dimorphic gene expression in the liver of male versus female mice (*Lau-Corona et al., 2017*; *Waxman et al., 1991*; *Zhang et al., 2012*; *Waxman and O'Connor, 2006*; *Norstedt and Palmiter, 1984*). In male mice, when GH signaling or secretory profile is disrupted, the expression of male-bias genes is dramatically suppressed and, concomitantly, the expression of female-bias genes are up-regulated due to their de-repression in male mice (*Lau-Corona et al., 2017*; *Waxman and O'Connor, 2006*; *Chia, 2014*). This is thought to be the result of altered male-bias (e.g. Bcl6) and female-bias (e.g. Cux2) transcription factor expression that reciprocally regulates the hepatic expression of sex-bias genes in male and female

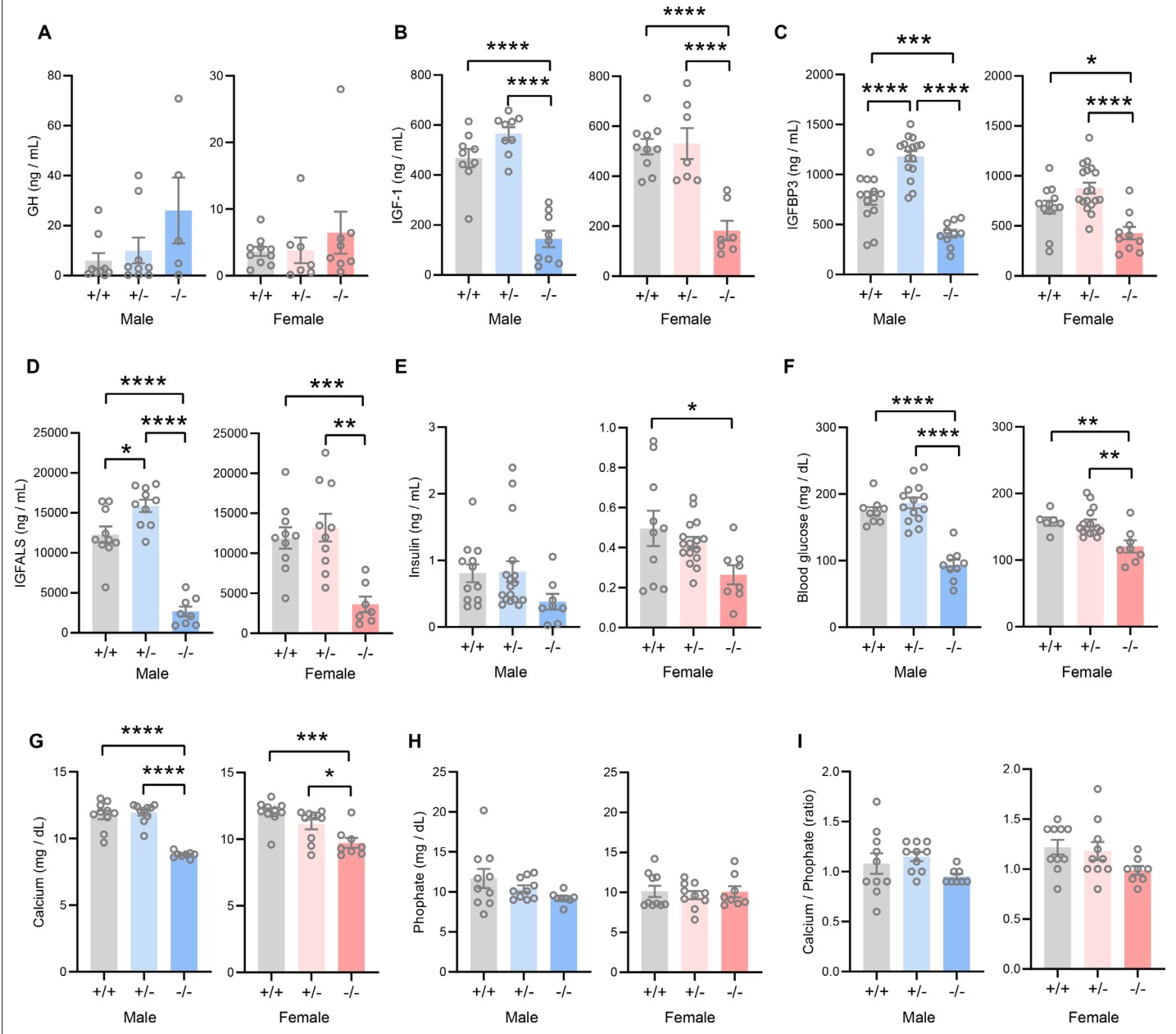

**Figure 4.** TMEM263 deficiency results in marked reduction in circulating insulin-like growth factor 1 (IGF-1), IGF binding protein 3 (IGFBP3), and IGF acid labile subunit (IGFALS) levels. Serum levels of growth hormone (GH; **A**), IGF-1 (**B**), IGFBP3 (**C**), IGFALS (**D**), insulin (**E**), glucose (**F**), calcium (**G**), and phosphate (**H**) in wild-type (WT) (+/+), heterozygous (+/-), and *Tmem263*-KO (-/-) male and female mice at 8 weeks of age. Sample size for panel A (GH): males (WT = 9; het = 9; KO = 5) and females (WT = 10; het = 7; KO = 8). Panel B (IGF-1): males (WT = 9; het = 9; KO = 9) and females (WT = 10; het = 7; KO = 7). Panel C (IGFBP3): males (WT = 15; het = 16; KO = 11) and females (WT = 12; het = 18; KO = 10). Panel D (IGFALS): males (WT = 10; het = 10; KO = 8) and females (WT = 10; het = 10; KO = 8). Panel E (insulin): males (WT = 12; het = 17; KO = 8) and females (WT = 10; het = 16; KO = 8). Panel F (glucose): males (WT = 9; het = 14; KO = 9) and females (WT = 6; het = 15; KO = 8). Panel G and H (calcium and phosphate): males (WT = 10; het = 10; KO = 8) and females (WT = 10; het = 10; KO = 8). (**I**) Ratio of calcium-to-phosphate in WT, heterozygous, and KO male and female mice. Sample size for males (WT = 10; het = 10; KO = 8) and females (WT = 10; het = 10; KO = 8). All data are presented as mean ± SEM. *p<0.05; **p<0.01; ***p<0.001; ****p<0.0001 (one-way ANOVA with Tukey's multiple comparisons test).

mice (**Conforto et al., 2012**; **Sugathan and Waxman, 2013**; **Meyer et al., 2009**). In male mice with disrupted GH signaling, *Bcl6* expression is suppressed and *Cux2* expression is markedly up-regulated, leading to transcriptional repression of male-bias genes and up-regulation of female-bias genes in the male liver (**Lau-Corona et al., 2017**). Remarkably, *Cux2* was one of the most up-regulated genes in

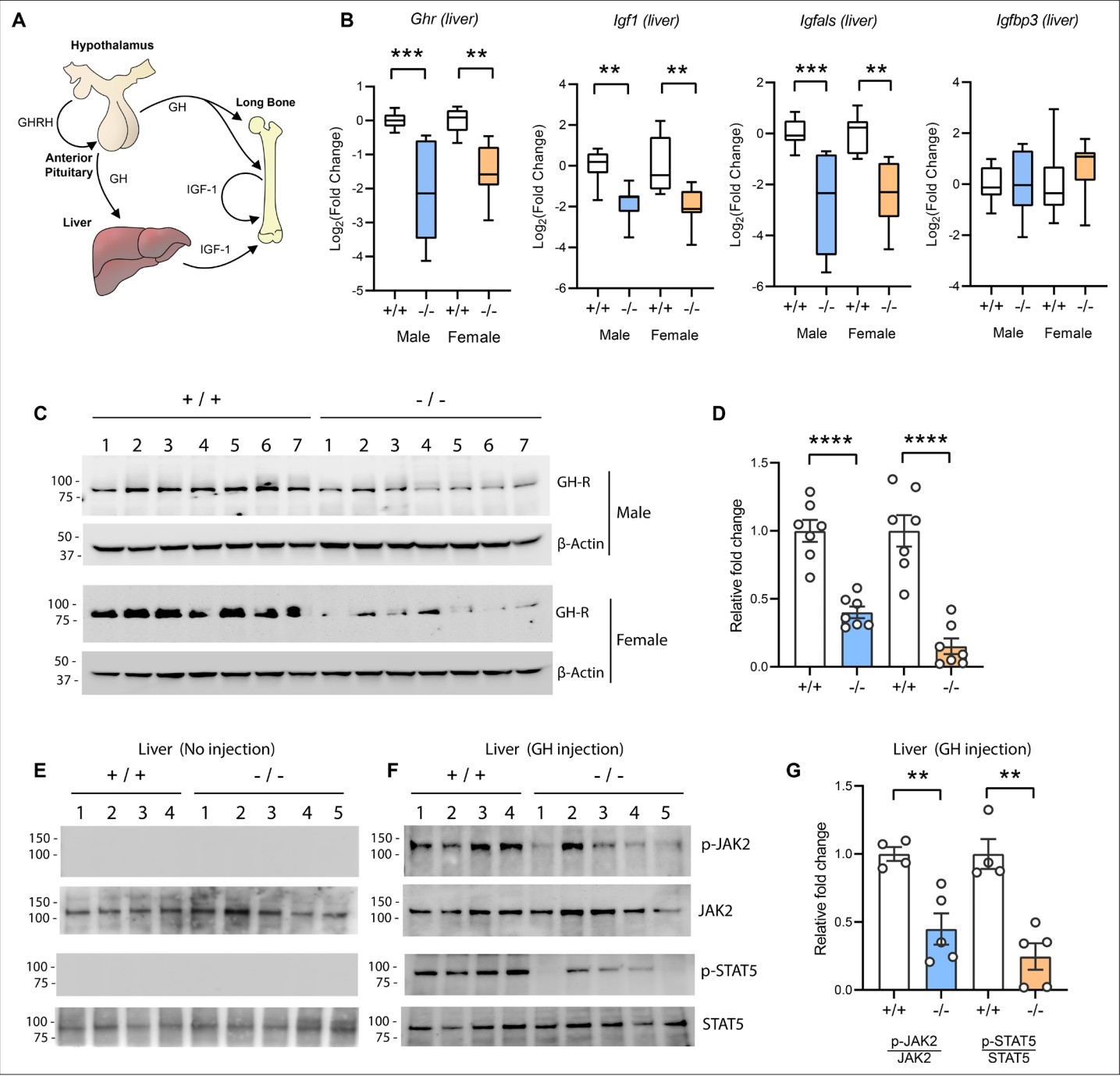

**Figure 5.** Reduced hepatic growth hormone receptor (GHR) protein level and signaling in TMEM263 knockout (KO) mice. (**A**) The growth hormone (GH)/insulin-like growth factor 1 (IGF-1) axis required for postnatal skeletal growth. At the onset of growth spurt, growth hormone releasing hormone (GHRH) from the hypothalamus causes the release of GH from the anterior pituitary. Circulating GH binds to its receptor (GHR) in liver and other peripheral tissues to induce the synthesis and secretion of IGF-1, which then acts in an endocrine, paracrine, and/or autocrine manner to induce skeletal growth. (**B**) Expression levels of *Ghr* (growth hormone receptor), *Igf1*, *Igfals* (IGF binding protein acid labile subunit), and *Igfbp3* (IGF binding protein 3) transcripts in the liver of wild-type (WT) and KO mice. Sample size of male mice (WT, n=8; KO, n=8) and female mice (WT, n=8; KO, n=8). (**C**) Immunoblot analysis of GHR protein levels in the liver of WT (n=7) and KO (n=7) mice. Molecular weight markers are indicated on the left. (**D**) Quantification of the immunoblot results as shown in C (n=7 per genotype). (**E–F**) Reduced hepatic GH-induced signaling in KO (-/-; n=5) mice relative to WT (+/+; n=4) controls. Immunoblot analysis of phospho-JAK2 (Tyr1008), total JAK2, phospho-STAT5 (Y694), and total STAT5 in liver lysates from control male mice not injected with GH (**E**) and male mice injected with recombinant GH (**F**). Molecular weight markers are indicated on the left of the gel. (**G**) Quantification of the immunoblot results as shown in F (WT, n=4; KO, n=5). All data are presented as mean ± SEM. **p<0.01; ***p<0.001; ****p<0.0001 (one-way ANOVA with Tukey's multiple comparisons test for data in B and two-tailed Student's *t*-test for data in D and F).

*Figure 5 continued on next page*

*Figure 5 continued*

The online version of this article includes the following source data for figure 5:

**Source data 1.** Top left – Original uncropped membrane from imager showing blue channel only as a black and white image.

**Source data 2.** Top left – Original uncropped membrane from imager showing blue channel only as a black and white image.

**Source data 3.** Top left – Original uncropped membrane from imager showing blue channel only as a black and white image.

general and was the most up-regulated transcription factor gene in the liver of *Tmem263*-KO male mice (*Figure 6C*). The expression of *Bcl6*, a male-bias transcription factor gene, was also significantly down-regulated in the KO male liver (*Figure 6C*). Consequently, all the well-known GH-regulated male-bias genes (growth and metabolism, cytochrome P450, and major urinary protein) were dramatically suppressed due to greatly diminished GH signaling, whereas the female-bias genes (e.g. distinct set of cytochrome P450) were concomitantly and markedly up-regulated due to *Cux2* overexpression in the male KO liver (*Figure 6D–F*).

To further demonstrate the feminization of the liver transcriptome of *Tmem263*-KO male mice, we compared our liver DEGs to three published datasets of mouse liver gene expression: (i) WT male vs female mice (*Conforto et al., 2012*), (ii) hypophysectomized vs sham control male mice (*Wauthier et al., 2010*), and (iii) *Stat5b*-KO vs WT male mice (*Clodfelter et al., 2006*). Overlap analysis was conducted to determine which DEGs are shared among these three groups and the *Tmem263*-KO male liver (*Figure 6G–I*). A complete list of all *Tmem263*-KO male liver DEGs that are also a DEG in at least one of the three published datasets is provided (*Figure 6—figure supplement 1*). When compared to WT male vs female liver data, *Tmem263*-KO male mice down-regulate 110 genes considered to be male-specific and up-regulate 174 genes considered to be female-specific (*Figure 6G*). This indicates in the male mouse, loss of Tmem263 results in a high degree of liver transcript feminization, a phenomenon usually prevented by normal GH signaling in male liver (*Lau-Corona et al., 2017*; *Waxman et al., 1991*; *Zhang et al., 2012*; *Waxman and O'Connor, 2006*; *Norstedt and Palmiter, 1984*). When compared to the DEG list of hypophysectomized male mouse liver – which lacks pituitary hormones (including GH) due to surgical removal of the pituitary gland – *Tmem263*-KO male liver displays 176 shared up-regulated genes and 128 shared down-regulated genes (*Figure 6H*). The high overlap of liver DEGs between *Tmem263*-KO and hypophysectomized male mice with a deficit in GH further suggests that Tmem263 deficiency disrupts GH action in the liver. Lastly, *Tmem263*-KO male mouse liver DEGs were compared to those of *Stat5b*-KO male mice (*Figure 6I*), which exhibit dwarfism and GH pulse resistance (*Udy et al., 1997*). We again observed a high degree of overlap between the two KO mouse models, as *Tmem263*-KO male liver shares 78 up-regulated and 91 down-regulated DEGs with *Stat5b*-KO male liver.

Altogether, these transcriptomic data provide supporting evidence that the underlying cause of the severe growth retardation and skeletal dysplasia seen in *Tmem263*-KO mice is most likely attributed to disrupted GH signaling in the liver and possibly other organs such as the bone (Figure *6J*).

## Discussion

*TMEM263* is a GWAS candidate gene associated with BMD (*Calabrese et al., 2017*). In this study, we used a genetic approach to functionally validate its causal role in postnatal growth. Mice lacking Tmem263 showed severe postnatal growth failure and had dramatically reduced bone volume and growth plate length. This phenotype is likely due to a disrupted GH/IGF-1 axis stemming from hepatic GHI and low IGF-1 production. This is the first study documenting the physiological function of this enigmatic, but highly conserved, membrane protein.

Since Laron's first description of dwarfism in humans due to defects in the GHR (*Eshet et al., 1984*; *Laron, 2004*; *Laron et al., 1966*), an enormous amount of progress has been made to understand how GH and IGF-1, as well as their binding proteins, receptors, and signaling pathways, contribute to linear growth (*David et al., 2011*; *Savage et al., 2011*; *Lanning and Carter-Su, 2006*; *Hwa et al., 2021*). In humans and animal models, defects in any component of the GH/IGF-1 axis leads to a variable severity of postnatal growth failure (short stature) and skeletal dysplasia. These include the initial discoveries of human mutations in GH (*Takahashi et al., 1996*; *Takahashi et al., 1997*; *Besson et al., 2005*), GHR (*Godowski et al., 1989*; *Amselem et al., 1989*; *Amselem et al., 1991*; *Berg et al.,*

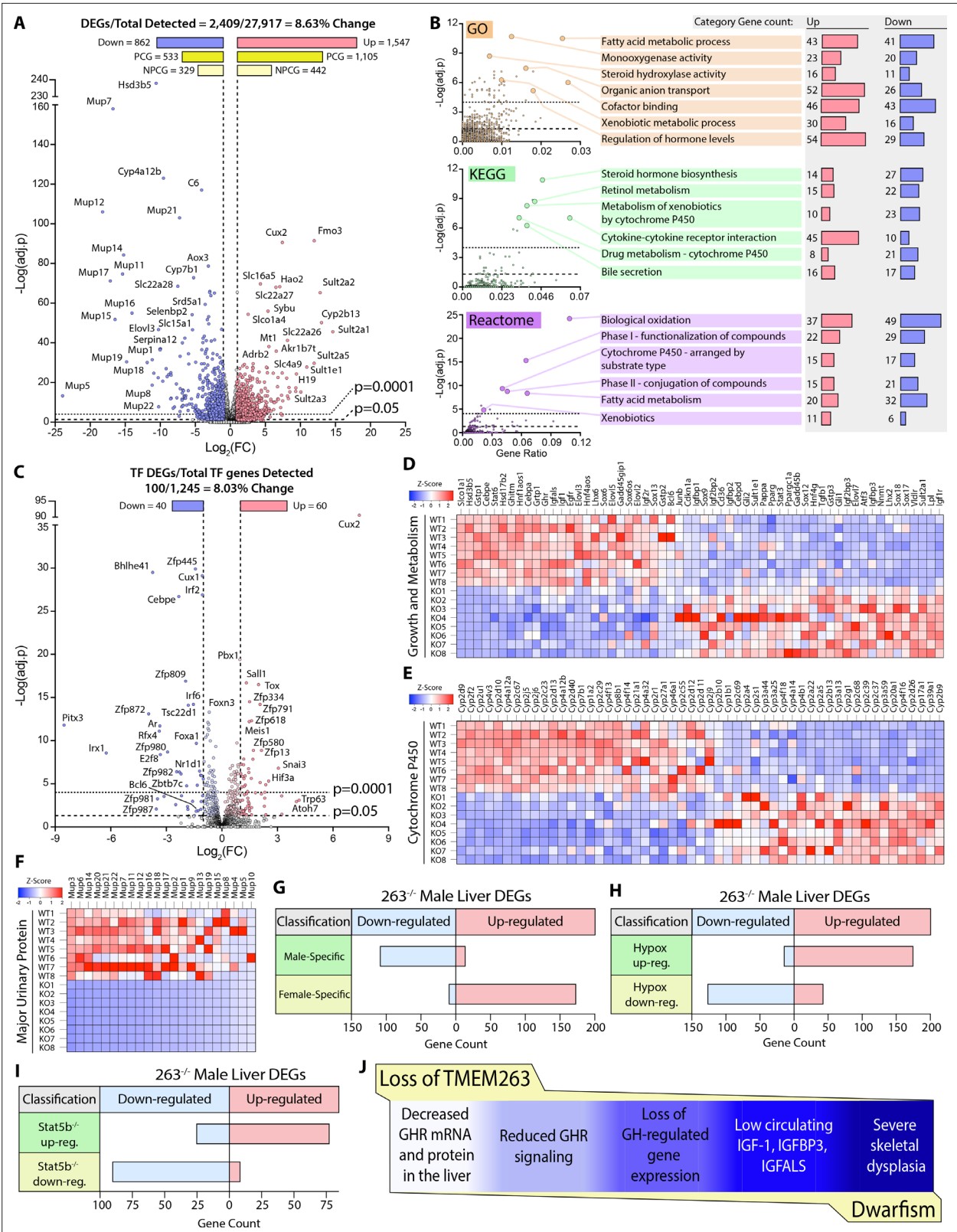

**Figure 6.** Loss of TMEM263 disrupts growth hormone (GH)-regulated gene expression in the male mouse liver. (**A**) Volcano plot of male mouse liver transcriptome. (**B**) Enrichment analysis of differentially expressed genes (DEGs) by Gene Ontology (GO), Kyoto Encyclopedia of Genes and Genomes (KEGG), and Reactome (http://www.reactome.org) databases. Plots of -Log(adj. p) vs gene ratio to show complete spread of all enrichment results for each analysis. The most effected categories are highlighted and labeled with the category name and number of up- and down-regulated genes

*Figure 6 continued on next page*

*Figure 6 continued*

within each. (**C**) Volcano plot showing the expression of all transcription factor (TF) genes detected in the male mouse liver transcriptome. TFs that are significantly up- or down-regulated in knockout (KO) male mouse liver are highlighted. (**D**) Heat map of DEGs involved in growth and metabolism. (**E**) Heat map of all protein-coding DEGs from the cytochrome P450 (Cyp) gene family. (**F**) Heat map of all protein-coding DEGs from the major urinary protein (Mup) gene family. (**G–I**) *Tmem263*-KO male liver DEG overlap comparison to three separate public datasets of mouse liver gene expression: (**G**) wild-type (WT) male vs. WT female mice (*Conforto et al., 2012*), (**H**) hypophysectomized vs. sham control male mice (*Wauthier et al., 2010*), and (**I**) *Stat5b*-KO vs WT male mice (*Clodfelter et al., 2006*). (**J**) Summary of key findings underpinning the dwarfism and skeletal dysplasia phenotypes of Tmem263-null mice. All heat map data is shown on a column z-score scale. Only significantly different genes (adjusted p-value <0.05 and Log$_2$(FC) <-1 or >1) are shown for all heat maps. PCG = protein-coding gene, NPCG = non-protein-coding gene. Sample size for male WT (n=8) and KO (n=8) mice (9-week-old) in all data shown.

The online version of this article includes the following figure supplement(s) for figure 6:

**Figure supplement 1.** All *Tmem263* knockout (KO) male mouse liver (263-/-) differentially expressed genes (DEGs) expressed by at least one other mouse model for comparison.

*1993*), IGF-1 (*Woods et al., 1996*), IGF-1 receptor (*Abuzzahab et al., 2003*; *Kawashima et al., 2005*; *Inagaki et al., 2007*), STAT5b (*Kofoed et al., 2003*; *Hwa et al., 2005*; *Vidarsdottir et al., 2006*), IGFALS (*Domené et al., 2004*; *Domené et al., 2007*), protein tyrosine phosphatase non-receptor type 11 (PTPN11/SHP-2) (*Tartaglia et al., 2001*; *Binder et al., 2005*), and pregnancy-associated plasma protein 2A (PAPPA2) (*Dauber et al., 2016*). The direct causal roles for these genes in regulating growth and skeletal acquisition have also been independently confirmed in genetic mouse models lacking one or more components of the GH/IGF-1 axis (*Efstratiadis, 1998*; *Qian et al., 2022*; *Lupu et al., 2001*; *Baker et al., 1993*; *Liu et al., 1993*). Because IGF-1 expression and secretion is directly regulated by GH (*Chia et al., 2010*; *Woelfle et al., 2003a*; *Woelfle et al., 2003b*), defects in the GH/IGF-1 axis would result in GHI, highlighted by the failure of endogenous or exogenously derived GH to elevate plasma IGF-1 levels (*David et al., 2011*; *Savage et al., 2011*; *Hwa et al., 2021*). In the clinical context, the cardinal biochemical features of GHI states are low circulating IGF-1 and normal or increased GH levels (*David et al., 2011*; *Savage et al., 2011*).

Interestingly, *Tmem263*-KO mice exhibited phenotypes closely resembling *Ghr* and *Igf1* KO mice (*Lupu et al., 2001*; *Zhou et al., 1997*; *Baker et al., 1993*) and humans with GHI. The main defect leading to the shortening of the growth plate in *Tmem263-KO* mice is decreased in the size of the proliferative zone, which was also reported in *Ghr* KO mice (*Sims et al., 2000*). Moreover, the dwarf *Tmem263-KO* mice had normal to marginally higher serum GH, but low IGF-1, IGFBP3, and IGFALS levels, which is also the case in *Ghr* KOs (*Zhou et al., 1997*). The critical proximal signaling events activated by GH binding to the GHR involves JAK2 kinase and the transcription factor STAT5b (*Lanning and Carter-Su, 2006*). Failure of STAT5b to bind to the cytoplasmic tails of the GHR due to a truncation mutation also leads to growth failure and a GHI state (*Milward et al., 2004*). In *Tmem263*-KO male mice, GH injection elicited a greatly muted JAK2/STAT5 signaling response relative to WT controls. GH is known to play an important role in regulating sex-dependent gene expression in liver, and a deficit in GH action results in the feminization of the male liver transcriptome (*Lau-Corona et al., 2017*; *Waxman et al., 1991*; *Zhang et al., 2012*; *Waxman and O'Connor, 2006*; *Norstedt and Palmiter, 1984*). Due to reduced hepatic GH signaling, the liver gene expression profile of *Tmem263*-KO male mice resembled that of WT female, hypophysectamized male, and *Stat5b*-KO male mice. This combination of phenotypes, along with a marked deficit in bone growth and dwarfism, suggests a likely defect in GHR signaling in *Tmem263*-KO mice. However, future mechanistic studies are needed to definitively establish the causal link between impaired GH/IGF-1 signaling and defect in skeletal acquisition and growth in *Tmem263*-KO mice.

What is the possible cause of GHI seen in *Tmem263*-KO mice? One explanation is the striking reduction in hepatic Ghr protein level induced by the loss of Tmem263. Because we also observed reduced expression of hepatic *Ghr* transcript, the decrease in Ghr protein level reflects, at least in part, lower transcription of the *Ghr* gene. Although reduced transcription could decrease *Ghr* mRNA level, other possible mechanisms could potentially contribute to the reduced *Ghr* transcript level as well; these include altered mRNA splicing, polyA adenylation, mRNA export, and mRNA stability. How a deficiency in Tmem263, a putative plasma membrane protein, led to altered hepatic *Ghr* transcript and protein level is presently unknown, and is a major limitation of the current study. We cannot, however, exclude the possibility that Tmem263 plays a posttranscriptional role in Ghr processing,

stability, turnover, trafficking, and/or signaling. Trafficking defect in GHR has been previously documented, where mutations located at the extracellular domain of the GHR failed to localize to the plasma membrane when expressed in cultured cells (*Duquesnoy et al., 1991*; *Maamra et al., 2006*; *Fang et al., 2008*). Whether these mutations affect protein folding and/or a transport signal recognized by another protein has not been established. One important question to address is whether TMEM263 physically interacts with GHR along the secretory pathway or on the plasma membrane of hepatocytes. If such physical association exists, it might provide mechanistic insights into how TMEM263 regulates GHR function. Since both human and mouse *TMEM263* and *GHR* are widely expressed across tissues (*Ballesteros et al., 2000*), as well as being highly expressed in liver (especially in human), the loss of Tmem263 in the KO mice may have a disproportionate impact on hepatic Ghr function. The generation and characterization of *Tmem263* cell-selective and tissue-specific KO mouse models will help unravel the impact of this protein across different organ systems.

It is not known whether the dwarf chicken with a loss-of-function mutation in *TMEM263* is also insensitive to GH stimulation. To date, there is only one documented case of a loss-of-function mutation in T*MEM263*, and it was associated with severe skeletal dysplasia in a fetus (*Mohajeri et al., 2021*). Because pregnancy was terminated before full term, it is unclear whether this mutation is compatible with postnatal survival and its impact on the severity of postnatal growth retardation. Nevertheless, it is tempting to speculate on the possibility that rare homozygous loss-of-function mutations and variants of TMEM263 may cause Laron-type dwarfism with characteristic GHI.

Intriguingly, only one homozygous missense mutation in the *TMEM263* gene has been discovered thus far from over 141,000 aggregated human exomes and genomes sequenced (Genome Aggregation Database; gnomAD v2.1.1) (*Karczewski et al., 2020*). Rare heterozygous splice-site, frameshift, and nonsense mutations in the *TMEM263* gene have also been documented in the gnomAD database (https://gnomad.broadinstitute.org/). Since the Asp-17 residue is highly conserved from fish to man, we speculate that the homozygous D17N missense mutation found in 40 individuals (allele frequency of 0.006795) may impact TMEM263 function and affect postnatal linear growth. It would be of considerable interest to ascertain whether individuals with the D17N variant have short stature. Since TMEM263 has not yet been associated with postnatal dwarfism in human, it should be considered a potential candidate gene in screening children with idiopathic growth failure.

# Materials and methods

**Key resources table**

| Reagent type (species) or resource | Designation | Source or reference | Identifiers | Additional information |
|---|---|---|---|---|
| Gene (*Mus musculus*) | Tmem263 | GenBank | Gene ID: 103266 | |
| Strain, strain background (*Mus musculus*) | Tmem263 knockout (-/-) mouse line; C57BL/6J genetic background | This paper | | The mouse line was generated at the Johns Hopkins Transgenic Core facility. The mouse line is available upon request |
| Cell line (*Homo sapiens*) | Human Embryonic Kidney cells (HEK 293T) | ATCC | Cat. #: CRL-3216 | Cell line has been authenticated by ATCC |
| Transfected construct (human) | Human TMEM263 expression plasmid in pCDNA3.1 vector backbone | Origene | Cat. #: (RC203933) | Contains a C-terminal Myc and FLAG epitope tag |
| Antibody | Anti-FLAG M2 (Mouse monoclonal) | Sigma | Cat. #: F1804 | WB (1:1000) |
| Antibody | Anti-JAK2 (Mouse monoclonal C-10) | Santa Cruz | Cat. #: sc-390539 | WB (1:100) |
| Antibody | Anti-STAT5 (Mouse monoclonal A-9) | Santa Cruz | Cat. #: sc-74442 | WB (1:100) |
| Antibody | Anti-GHR (Mouse monoclonal B-12) | Santa Cruz | Cat. #: sc-137184 | WB (1:100) |
| Antibody | Anti-phospho-JAK2 (Tyr1008) (Rabbit monoclonal D4A4) | Cell Signaling Technology | Cat. #: sc-137184 | WB (1:500) |

*Continued on next page*

*Continued*

| Reagent type (species) or resource | Designation | Source or reference | Identifiers | Additional information |
|---|---|---|---|---|
| Antibody | Anti-phospho-STAT5 (Tyr694) (Rabbit polyclonal) | Cell Signaling Technology | Cat. #: sc-137184 | WB (1:500) |
| Antibody | Anti-β-Actin (Mouse monoclonal) | Sigma | Cat. #: A1978 | WB (1:2000) |
| Antibody | Anti-TMEM263 (Rabbit polyclonal) | Origene | Cat. #: TA333490 | WB (1:1000) |
| Peptide, recombinant protein | Recombinant human growth hormone | PeproTech | Cat. #: 100-40 | Dose injected into mice (3 µg/g body weight) |
| Commercial assay or kit | Pierce Cell Surface Isolation kit | Thermo Fisher Scientific | Cat. #: 89881 | |
| Commercial assay or kit | Insulin ELISA kit | Crystal Chem | Cat. #: 90080 | |
| Commercial assay or kit | Growth hormone (GH) ELISA kit | Millipore Sigma | Cat. #: EZRMGH-45K | |
| Commercial assay or kit | IGF1 ELISA kit | Crystal Chem | Cat. #: 80574 | |
| Commercial assay or kit | IGFBP3 ELISA kit | Abcam | Cat. #: ab100692 | |
| Commercial assay or kit | IGFALS ELISA kit | Cusabio | Cat. #: CSB-EL011094MO | |
| Commercial assay or kit | iScript cDNA synthesis kit | Bio-Rad | Cat. #: 1708891 | |
| Commercial assay or kit | iTaq Universal SYBR Green Supermix | Bio-Rad | Cat. #: 1725124 | |
| Chemical compound | Trizol Reagent | Thermo Fisher Scientific | Cat. #: 15596018 | |

## Mouse model

The *Tmem263* KO mouse strain was generated at the Johns Hopkins University School of Medicine Transgenic Core facility. To obtain a *Tmem263*-null allele, exon 3 that encodes ~81% of the full-length protein (GenBank # NP_689474) was deleted with the CRISPR/Cas9 method. The deletion encompasses amino acid 23–115. The two guide RNAs (gRNA) used were 5′-CTATCATATGTCTCTCATTAG-3′ and 5′-GCCACTGGCATCA TATGCTG-3′. The *Tmem263* KO mice were generated and maintained on a C57BL/6J genetic background. Genotyping primers for *Tmem263* WT allele were forward (263-F2) 5′-GCAAGAGCTCCTTACTTAC TCAG-3′ and reverse (263-R1) 5′- GATAAGGGCACTTTGTTTACAACT G-3′. The size of the WT band was 391 bp. Genotyping primers for the *Tmem263* KO allele were forward (263-F2) 5′-GCAAGAGCTCCTTACTTACTCA G-3′ and reverse (263-R2) 5′-TCACCAATACTT TCAACACAGCAG-3′. The size of the KO band was 218 bp. The GoTaq Green Master Mix (Promega, M7123) was used for PCR genotyping. The genotyping PCR parameters were as follows: 94°C for 5 min, followed by 10 cycles of (94°C for 10 s, 65°C for 15 s, 72°C for 30 s), then 25 cycles of (94°C for 10 s, 55°C for 15 s, 72°C for 30 s), and lastly 72°C for 5 min. Genotyping PCR products from WT and KO mice were excised and confirmed by DNA sequencing. All mice were generated by intercrossing *Tmem263* heterozygous (+/-) mice. *Tmem263* WT (+/+), heterozygous (+/-), and KO (-/-) littermates were housed in polycarbonate cages on a 12 hr light-dark photocycle with ad libitum access to water and food. Mice were fed a standard chow (Envigo; 2018SX). At termination of the study, all mice were fasted for 1–2 hr and euthanized. Tissues were collected, snap-frozen in liquid nitrogen, and kept at –80°C until analysis. All mouse protocols (protocol # MO22M367) were approved by the Institutional Animal Care and Use Committee of the Johns Hopkins University School of Medicine. All animal experiments were conducted in accordance with the National Institute of Health guidelines and followed the standards established by the Animal Welfare Acts.

## Expression plasmid and antibodies

Mammalian expression plasmid encoding human TMEM263 with a C-terminal epitope tag (Myc-DDK) was obtained from Origene (RC203933). Control pCDNA3.1 empty plasmid was obtained from Invitrogen. Rabbit polyclonal anti-human C12orf23 (TMEM263) was from Origene (TA333490). Mouse

monoclonal JAK2 (C-10), STAT5 (A-9), growth hormone receptor (GHR) (B-12) were obtained from Santa Cruz (sc-390539, sc-74442, and sc-137184, respectively). Rabbit monoclonal antibody against phospho-JAK2 (Tyr1008)(D4A8) was obtained from Cell Signaling Technology (CST # 8082). Rabbit polyclonal antibody against phospho-STAT5 (Y694) was obtained from Cell Signaling Technology (CST#9351). Mouse monoclonal anti-FLAG M2 antibody (F1804) and mouse monoclonal anti-β-actin antibody (A1978) were obtained from Sigma. HRP-conjugated goat anti-rabbit (CST#7074) and goal anti-mouse (CST#7076) antibodies were from Cell Signaling Technology.

## Cell surface biotinylation

Cell surface proteins from transfected HEK293 cells were biotinylated using the Pierce Cell Surface Protein Isolation Kit (Thermo Fisher Scientific, cat. #: 89881). Briefly, transfected cells were washed with ice-cold PBS, then incubated with the cell-impermeable biotinylation reagent (Sulfo-NHS-SS-Biotin) for 30 min at 4°C to prevent endocytosis. A quenching solution was then used to stop biotinylation. Cells were collected in a clean tube and centrifuged at 500× $g$ for 3 min. The cell pellet was then lysed with RIPA buffer (10 mM Tris-HCl pH 8.0, 1 mM EDTA, 1% Triton X-100, 0.1% sodium deoxycholate, 0.1% sodium dodecyl sulfate, 140 mM NaCl, 1 mM PMSF) supplemented with protease inhibitor cocktail (Roche, 11836153001) for 30 min on ice. Cell lysate was then centrifuged at 10,000 × $g$ for 2 min at 4°C and the resultant supernatant was collected. Biotinylated proteins were pulled down with Pierce NeutrAvidin Agarose (Thermo Fisher Scientific; 29200) at room temperature for 1 hr mixed end-over-end. Bound proteins were eluted from the agarose by DTT (50 mM, final) and heat (94°C for 3 min). Fractions of the input cell lysate, flow-through/non-bound solution, and eluate were used for immunoblot analysis.

## Skeletal phenotyping

For Alcian blue/Alizarin red stained whole mounts, skeletal preps were preformed using P0 neonates following established protocols (*Rigueur and Lyons, 2014*). Briefly, carcasses were macerated by removing skin, eyes, internal organs, and adipose tissue before dehydrating in 95% ethanol overnight and then fixing in acetone. Cartilage was stained by submerging in Alcian blue solution overnight at room temperature. Mineralized tissue was stained by submerging in Alizarin red solution at 4°C overnight.

For microcomputed tomographic analyses of bone structure, high-resolution images of the mouse femur were acquired using a desktop microtomographic imaging system (Skyscan 1172, Bruker, Belgium) in accordance with the recommendation of the American Society for Bone and Mineral Research (*Bouxsein et al., 2010*). Samples were scanned with an isotropic voxel size of 10 μm at 65 keV and 153 μA using a 1.0 mm aluminum filter. The resulting images were reconstructed using NRecon (Bruker). Trabecular bone parameters in the distal femur of control mice were assessed in an ROI 500 μm proximal to the growth plate and extending for 2 mm, while femoral cortical bone parameters were assessed in a 500 μm ROI centered on the mid-diaphysis. Due to the reduced size of the *Tmem263*-KO mice, ROI were adjusted to be proportional for tissue length. Trabecular bone parameters were assessed in an ROI 300 μm proximal to the growth plate and extending for 1.2 mm; cortical bone parameters were assessed in a 300 μm ROI centered on the mid-diaphysis.

Growth plate morphology was examined in the proximal tibia. Tibia were dehydrated in graded alcohol solutions and then embedded in methyl methacrylate. 7 μm thin sections were cut with a Leica HistoCore Automated Rotary microtome (Leica, Wetzlar, Germany) and stained with Goldner's modified trichrome. Total growth plate length, as well as the lengths of the hypertrophic and proliferative zones were measured with Bioquant Image Analysis software (Bioquant, Nashville, TN, USA).

## Western blot analysis

Liver protein was isolated using RIPA buffer as previously described (*Rodriguez et al., 2016*). Liver protein lysates used for immunoblots were boiled for 5 min in a loading buffer (50 mM Tris, 2% SDS, 1% β-ME, 6% glycerol, 0.01% bromophenol blue). Total protein was quantified by BCA assay (Thermo Scientific, 23225), loaded in equal amounts and volume, and run on a 7.5% or 10% TGX gel (Bio-Rad, 4561023 and 4561033). Protein was transferred to PVDF membrane (Bio-Rad, 1620177) using the Trans Blot Turbo system (Bio-Rad) and blocked in PBS containing 0.2% Tween 20 and 5% non-fat milk for 1 hr, then incubated overnight at 4°C on a shaker with primary antibody. The following primary

antibodies were used: anti-TMEM263 (1:1000), anti-GH-R (1:100), anti-phospho-JAK2 (Tyr1008) (1:500), anti-phospho-STAT5 (Y694) (1:500), anti-JAK2 (1:100), anti-JAK5 (1:100), anti-β-actin (1:2000). Blots were washed at least three times (5 min each) with PBS containing 0.2% Tween 20 before secondary antibody was added. HRP-conjugated anti-rabbit or anti-mouse secondary antibody (1:500 to 1:1000; 1 hr at room temperature) was used to recognize the primary antibody. Blots were washed at least three times (5 min each) with PBS containing 0.2% Tween 20. Immunoblots were developed using HRP substrate ECL (GE Healthcare), visualized with a MultiImage III FluorChem Q (Alpha Innotech), and quantified with ImageJ (*Schneider et al., 2012*).

## GH injection

Recombinant human GH from Peprotech (Cat. #: 100-40; Cranbury, NJ, USA) was reconstituted in PBS at a concentration 0.2 µg/µL. Each mouse received GH injection at a dose of 3 µg/g body weight. Mice were sacrificed 25 min post GH injection, and the liver was immediately harvested and snap-frozen in liquid nitrogen.

## Blood and tissue chemistry analysis

Tail vein blood samples were allowed to clot on ice and then centrifuged for 10 min at 10,000 × *g*. Serum samples were stored at –80°C until analyzed. Serum insulin (Crystal Chem, Elk Grove Village, IL, USA; 90080), GH (Millipore Sigma; EZRMGH-45K, Burlington, MA, USA), IGF-1 (Crystal Chem, 80574), IGFBP3 (Abcam; ab100692, Waltham, MA, USA), and IGFALS (Cusabio; CSB-EL011094MO, Houston, TX, USA) levels were measured by ELISA according to the manufacturer's instructions. Serum calcium and phosphate levels were quantified at the Molecular and Comparative Pathobiology Core at The Johns Hopkins University School of Medicine using the Respons910VET Veterinary Chemistry Analyzer (DiaSys Diagnostic Systems, Wixom, MI, USA).

## Quantitative real-time PCR analysis

Total RNA was isolated from tissues using Trizol reagent (Thermo Fisher Scientific) according to the manufacturer's instructions. Purified RNA was reverse-transcribed using an iScript cDNA Synthesis Kit (Bio-Rad). Real-time quantitative PCR analysis was performed on a CFX Connect Real-Time System (Bio-Rad) using iTaq Universal SYBR Green Supermix (Bio-Rad) per manufacturer's instructions. Data were normalized to either *β-actin* or *36B4* gene (encoding the acidic ribosomal phosphoprotein P0) and expressed as relative mRNA levels using the ΔΔCt method (*Schmittgen and Livak, 2008*). Fold change data were log transformed to ensure normal distribution and statistics were performed. Real-time qPCR primers used were: mouse *Tmem263* forward (qPCR-263-F1), 5'-CGCGGTGATCAT GAATCAGGCAG-3' and reverse (qPCR-263-R1), 5'-GCTCCCTTTGTAACACTGAAGA-3'; growth hormone receptor (*Ghr*) forward, 5'-ACAGTGCCTACTTTTGTGAGTC-3' and reverse, 5'-GTAGTGGT AAGGCTTTCTGTGG-3'; *Igf1* forward, 5'-GTGAGCCAAAGACACACCCA-3' and reverse, 5'-ACCT CTGATTTTCCGAGTTGC-3'; *Igfbp3* forward, 5'-CCAGGAAACATCAGTGAG TCC-3' and reverse, 5'- GGATGGAACTTGGAATCGGTCA-3'; *Igfals* forward, 5'-CTGCCCGATAGCATCCCAG-3' and reverse, 5'-GAAGCCAGACTTGGTGTGTGT-3'; *36B4* forward, 5'-AGATTCGGGGATATGCTGTTGGC-3' and reverse, 5'-TCGGGTCCTAGACCAGTGTTC-3'; β-actin forward, 5'-GGCACCACACCTTCTACAATG-3' and reverse, 5'-GGGGTGTTGAAGGTCTCAAAC-3'.

## RNA sequencing and bioinformatics analysis

Bulk RNA sequencing of WT (n=8) and *Tmem263*-KO (n=8) mouse liver were performed by Novogene (Sacramento, CA, USA) on an Illumina platform (NovaSeq 6000) and pair-end reads were generated. Sequencing data was analyzed using the Novogene Analysis Pipeline. In brief, data analysis was performed using a combination of programs, including Fastp, Hisat2, and Feature-Counts. Differential expressions were determined through DESeq2. The resulting p-values were adjusted using the Benjamini and Hochberg's approach for controlling the false discovery rate. Genes with an adjusted p-value ≤0.05 found by DESeq2 were assigned as differentially expressed. Gene Ontology, Kyoto Encyclopedia of Genes and Genomes, and Reactome (http://www.reactome.org) enrichment were implemented by ClusterProfiler. All volcano plots and heat maps were generated in GraphPad Prism 9 software. All statistics were performed on log transformed data. All heat maps were generated from column z-score transformed data. The z-score of each column

was determined by taking the column average, subtracting each sample's individual expression value by said average then dividing that difference by the column standard deviation. Z-score = (value - column average)/column standard deviation. High-throughput sequencing data from this study have been submitted to the NCBI Sequence Read Archive (SRA) under accession number # PRJNA938158.

## Statistical analyses

All results are expressed as mean ± standard error of the mean (SEM). Statistical analysis was performed with GraphPad Prism 9 software (GraphPad Software, San Diego, CA, USA). Data were analyzed with two-tailed Student's $t$-tests or by one-way ANOVA with Tukey's multiple comparisons test. $p<0.05$ was considered statistically significant.

## Acknowledgements

This work was supported by the National Institutes of Health (DK084171 to GWW; AR077533 and DK099134 to RCR). We thank Thomas Clemens for helpful discussion.

## Additional information

### Funding

| Funder | Grant reference number | Author |
|---|---|---|
| National Institute of Diabetes and Digestive and Kidney Diseases | DK084171 | G William Wong |
| National Institute of Diabetes and Digestive and Kidney Diseases | DK099134 | Ryan C Riddle |
| National Institute of Arthritis and Musculoskeletal and Skin Diseases | AR077533 | Ryan C Riddle |

The funders had no role in study design, data collection and interpretation, or the decision to submit the work for publication.

### Author contributions

Dylan C Sarver, Conceptualization, Formal analysis, Investigation, Visualization, Writing - review and editing; Jean Garcia-Diaz, Investigation, Methodology; Muzna Saqib, Investigation, Writing - review and editing; Ryan C Riddle, Formal analysis, Supervision, Investigation, Visualization, Writing - review and editing; G William Wong, Conceptualization, Formal analysis, Supervision, Funding acquisition, Investigation, Writing – original draft, Project administration

### Author ORCIDs

Ryan C Riddle (ID) http://orcid.org/0000-0001-7265-6939
G William Wong (ID) https://orcid.org/0000-0002-5286-6506

### Ethics

All mouse protocols (protocol # MO22M367) were approved by the Institutional Animal Care and Use Committee of the Johns Hopkins University School of Medicine. All animal experiments were conducted in accordance with the National Institute of Health guidelines and followed the standards established by the Animal Welfare Acts.

Reviewer #2 (Public Review): https://doi.org/10.7554/eLife.90949.3.sa1
Reviewer #3 (Public Review): https://doi.org/10.7554/eLife.90949.3.sa2
Author Response https://doi.org/10.7554/eLife.90949.3.sa3

## Additional files

### Supplementary files
• MDAR checklist

### Data availability
High-throughput sequencing data from this study have been submitted to the NCBI Sequence Read Archive (SRA) under accession number # PRJNA938158.

The following dataset was generated:

| Author(s) | Year | Dataset title | Dataset URL | Database and Identifier |
|---|---|---|---|---|
| Sarver DC, Garcia-Diaz J, Saqib M, Riddle RC, Wong GW | 2023 | *Mus musculus* strain:C57BL/6J (house mouse) | https://www.ncbi.nlm.nih.gov/bioproject/?term=PRJNA938158 | NCBI BioProject, PRJNA938158 |

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
