## [Editor Report · eLife assessment]

This study discloses **important** physiological function for TMEM63 in regulating postnatal growth in mice. The data supporting the impaired body growth and skeletal phenotype as well as disrupted growth hormone/insulin-like growth factor-I (GH/IGF-I) signaling in TMEM63 knockout mice are **compelling**. However, to establish that alteration of hepatic GH/IGF-I signaling is the cause for observed growth and skeletal phenotype in TMEM63 knockout mice would need additional work.

---

## [Referee Report · Reviewer #2 (Public Review)]

Summary:

The study demonstrates that deletion of a small cytoplasmic membrane protein, Tmem263, caused severe impairment of longitudinal bone growth and that the impaired bone growth was caused by suppression of expression and/or protein levels of growth hormone receptor in the liver.

Strengths:

The experimental design of the study is sound and the results are in general of supportive of the conclusions.

Weaknesses:

The study lacks mechanistic investigation into how the deletion of a gene corresponding to a small cytoplasmic membrane protein would lead to substantial reduction in the gene expression of growth hormone receptor, which takes place in the nuclei. Accordingly, the manuscript is of largely descriptive nature.

---

## [Referee Report · Reviewer #3 (Public Review)]

Prior studies in humans and in chickens suggested that TMEM263 could play an important role in longitudinal bone growth, but a definitive assessment of the function and potential mechanism of action of this species-conserved plasma membrane protein has been lacking. Here, the authors create a TMEM263 null mouse model and convincingly show dramatic cessation of post-natal growth, which becomes apparent by day PND21. They report proportional dwarfism, highly significant bone and related phenotypes, as well as notable alterations of hepatic GH signaling to IGF1. A large body of prior work has established an essential role for GH and it's stimulation of IGF1 production in liver and other tissues in post-natal growth. On this basis, the authors conclude that the observed decrease in serum IGF1 seen in TMEM263-KO mice is causal for the growth phenotype, which seems likely. Moreover, they ascribe the low serum IGF1 to the observed decreases in hepatic GH receptor (GHR) expression and GHR/JAK2/STAT5 signaling to IGF1, which is plausible but not proven by the experiments presented.

The finding that TMEM263 is essential for normal hepatic GHR/IGF1 signaling is an important, and unexpected finding, one that is likely to stimulate further research into the underlying mechanisms of TMEM263 action, including the distinct possibility that these effects involve direct protein-protein interactions between GHR and TMEM263 on the plasma membrane of hepatocytes, and perhaps on other mouse cell types and tissues as well, where TMEM263 expression is up to 100-fold higher (Fig. 1C).

An intriguing finding of this study, which is under emphasized and should be noted in the Abstract, is the apparent feminization of liver gene expression in male TMEM263-KO mice, where many male-biased genes are downregulated, and many female-biased genes are upregulated. Further investigation of these liver gene responses by comparison to public datasets could be very useful, as it could help determine: (1) whether the TMEM263 liver phenotype is similar to that of hypophysectomized male mouse liver, where GH and GHR/STAT5/IGF1 signaling are both totally ablated; or alternatively, (2) whether the phenotype is more similar to that of a male mouse given GH as a continuous infusion, which induces widespread feminization of gene expression in the liver, and is perhaps similar to the gene responses seen in the TMEM263-KO mice. Answering this question could provide critical insight into the mechanistic basis for the hepatic effects of TMEM263 gene KO.

Comments on revised version:

The authors have addressed a majority of the concerns raised during the initial review. The evidence supporting the whole-body growth and skeletal phenotypes, as well as the disruption of GH/IGF1 signaling seen in TMEM263-KO mice, is convincing. However, there is insufficient evidence to definitively conclude that the observed alteration of hepatic GH/IGF1 signaling is causative of the body growth and skeletal phenotypes.

---

## [Author Response]

The following is the authors’ response to the original reviews.

Based on the reviewer comments (see below) and subsequent discussion between the reviewers and the Reviewing Editor, I would like to invite the authors to make major revisions, including new experiments. However, if major new experiments are not feasible, as may be the case, then at a minimum, I would urge the authors to:

1. Tone down the language regarding a causative role for changes in GH/IGF-I signaling in mediating the effects of Tmem63 on the skeleton, and also be very open in acknowledging the lack of mechanistic insight into how Tmem regulates GH signaling.

Response: We toned down the language as suggested and also acknowledged the lack of mechanistic insights into how Tmem263 regulates GH signaling.

1. Revise/redo or if not possible, then delete the problematic experiment in Fig. 5E.

Response: We have included additional Western blot data in Figure 5 from control WT and KO male mice without exogenous GH injection. In the absence of GH injection, we could not detect Jak2 and Stat5 phosphorylation in the liver of male WT and KO mice.

1. Address the comments about liver feminization.

Response: We have performed additional analysis as suggested by reviewer # 3. We have now included additional data to address the issue of liver feminization (new Fig. 6G-I and Figure 6-figure supplement 1). We plan to expand on this very topic in future studies as this is an interesting transcriptional phenomenon.

1. Revise the manuscript to address as many of the recommendations for the authors as possible, many of which can be addressed by textual edits.Response: We have addressed as many of the textual changes as suggested in the revised manuscript.

**Reviewer #2 (Recommendations for The Authors):**
TMEM263 has been suggested to be associated with bone mineral density and growth in humans and mice, but the functional role of this transmembrane protein in the regulation of bone metabolism is unknown. With the knockout mouse approach, this manuscript demonstrates that Tmem263 is essential for longitudinal bone growth in the mouse as deletion of Tmem263 in knockout (KO) mice developed severe postnatal growth impairment and proportional dwarfism. It is determined that the dwarfism was caused by a substantial reduction in liver expression of growth hormone receptor (GHR), a slight increase in serum GH, and a reduction in serum IGF-I, which resulted in disruptive of GH/IGF-I regulatory axis of endochondral bone formation.The study was relatively well designed, and the results in general are supportive of the conclusions. While this study discloses new and intriguing functional information about a novel cytoplasmic membrane gene, there are a few minor issues that the authors may wish to address. These issues are listed in the following:1. One of the intriguing findings of this manuscript is that deletion of a gene encoding a small cytoplasmic membrane protein could cause a substantial reduction in the expression and protein levels of GHR. Inasmuch as a couple of potential explanations were offered in the Discussion section (first complete paragraph of page 10), there has been no attempt to test any of the suggested causes, since many of these potential mechanisms can readily be tested experimentally. Accordingly, the lack of mechanistic investigation into this intriguing effect renders the manuscript largely descriptive in nature.

Response: The point made by the reviewer is well taken. We do plan to have follow up studies to establish which among the mechanisms we highlighted in the discussion is contributing to the reduction in GHR transcript and protein level. Our present study is the first functional characterization of this enigmatic novel membrane protein. We anticipate that multiple follow-up studies are needed to gain a deeper understanding of the biology of Tmem263. We believe that our present study represents an important first step.

1. Because a major conclusion is that the bone phenotype of Tmem263 KO mice was caused by deficient hepatic expression and/or action of GHR, it would be helpful to (or strengthen) the conclusion if a brief comparison of the bone phenotype between GHR KO mice and Tmem263 KO mice is included in the Discussion section.

Response: We have now included this information in the revised manuscript.

1. In Figure 3, the cortical bone parameters (i.e., Tt.Ar, Ct.Ar, and Ct.Th), but none of the trabecular bone parameters (i.e., BV/TV, Tb.N, Tb.Th), were normalized against femur length. The authors did not provide a rationale for this differential treatment with the cortical bone parameters from the trabecular bone parameters. If the reason to normalize the cortical bone parameters against bone length was to demonstrate that the reduced cortical bone mass in mutants was related to the impaired longitudinal bone growth, then why did the authors not also assess whether the observed reduction in these trabecular bone parameters in KO mutants was proportional to reduced longitudinal bone growth?

Response: We actually made the exact adjustments that the reviewer refers to, as stated in the methods section. Please see page 14. The regions of interest (ROIs) of both the trabecular bone analysis and the cortical analysis in the mutants was reduced proportional to the length of the bone (40% smaller). The normalization to Tt Ar to femur length in Figure 3I was only meant to show that the reduction in Tt Ar in the mutants was proportional. We have modified the text in our result section for clarity.

1. Elements described in Fig. 5A have been well documented. Therefore, Fig. 5A is unnecessary and can be deleted.

Response: We felt that Figure 5A should remain. It helps orient readers that are not familiar with the literature to be aware that both liver- and bone-derived IGF-1 contribute to longitudinal bone growth.

1. Figure 6 was performed with male KO mice. Were the altered gene expression profiles in female KO mice any different from male KO mice?

Response: We plan to perform RNA-seq in female mouse liver in our follow-up studies. We do not know, at present, whether and to what extent the liver transcriptomic profile would be different between male and female KO mice. As far as dwarfism and deficiency in skeletal acquisition, both male and female KO mice showed the same phenotypes.

1. The number of animals (or samples) per group in some of the Figures (i.e., Fig. 2G, 2I, 2J, 3A to J, the entire Fig. 4, 5D, 5F, and Suppl Fig. 1) is needed to be provided in the legends.

Response: We have included this information in the figure legends.

**Reviewer #3 (Recommendations for The Authors):**
1. Explain the discrepancy between the impact of KO on serum Igfbp3 ( = decreased) vs. hepatic Igfbp3 ( = unchanged).

Response: We do not have a plausible mechanism, at present, that can explain the reduction in circulating serum Igfbp3 level without an apparent reduction in Igfbp3 transcript level in the liver. In human studies, typically only serum IGFBP3 levels are measured but not the hepatic IGFBP3 transcript level. Therefore, it is unclear whether the circulating levels of IGFBP3 is being regulated at the posttranscriptional level, an issue that can be explored in future studies.

1. Line 215, 221, and elsewhere - Foxa1 does not show significant male-biased expression in mouse liver.

Response: We have removed Foxa1 from the text.

1. Line 225- According to the abstract of Ref. #45, Cux2 regulates a subset of sex-biased genes in the liver. The authors should compare the genes dysregulated by TMEM263-KO (Fig. 6) to those altered by Cux2 loss (Ref. #45) to ascertain whether the results of Fig. 6 are partially or entirely explained by Cux2 overexpression.

Response: We agree that this is a great area of future study. We do feel this, however, would be better explored in a more in-depth follow-up article. We felt, given the current direction of the paper it made more sense to include differential expression comparisons of male vs female, hypophysectomized vs sham control, and Stat5b-KO vs WT mouse liver gene expression data. Our future work will explore the transcriptomes of male and female WT and Tmem263-KO liver gene expression in the context of the observed physiology.

1. Line 262- "lower transcription of Ghr gene". A decrease in mRNA levels does NOT equate with a decrease in transcription per se. Altered mRNA splicing, poly A, export, cytoplasmic stability, etc. are all potential contributors.

Response: We have included these possibilities highlighted by the reviewer in our revised Discussion section.

1. Line 273, "TMEM263... most highly expressed in liver" Not correct - see Fig. 1C for TMEM263 RNA levels in mouse tissues.

Response: We have corrected the text on page 11.

1. Line 425 - Include GEO accession number.

Response: We have already uploaded our RNA-seq data to the NCBI Sequence Read Archive (SRA), and the data can be accessed under accession number # PRJNA938158.

1. Fig. 6 - Line 796 - Specify the age and sex of mice analyzed.

Response: We have included the information in the revised figure 6 legend.

1. Fig.2 - Suppl 1- Specify age of mice.

Response: We have included the information in the revised Figure 2-figure supplement 2.

1. Fig.2G -Specify the sex of the mice.

Response: For the P1 to P21 pups’ data, we did not separate by sex, as gender determination of pups at P1 and P7 can be challenging. We now indicated this in the figure legend.

1. Fig. 6A and 6C-6F: Which of these genes shows sex-dependent expression in wild-type liver? Use color to highlight gene names for genes that show male-biased or female-biased expression.

Response: We agree with the reviewer that additional labels on Figure 6A and 6C-F would be helpful to show genes of sex-bias. However, this is not the primary point of the paper. This topic deserves a much more in-depth analysis in follow up studies focused on defining the exact type and degree of transcript feminization in the liver of Tmem263-KO mice, as well as, its physiologic consequences. For readers interested in this topic, we have included the subfigures G-I in Figure 6 and for greater transcript level detail, figure 6 supplement 1.